# STRATEGICALLY-LINKED DECISIONS IN LONG-TERM PLANNING AND REINFORCEMENT LEARNING

## ABSTRACT

Long-term planning, as in reinforcement learning (RL), is often hard to interpret as it involves strategies: collections of actions that work toward a goal with potentially complex dependencies. In particular, some actions are taken at the expense of short-term benefit to enable future actions with even greater returns. In this paper, we quantify such dependencies between planned actions with *strategic link scores*: the drop in the likelihood of an earlier action under the constraint that a follow-up action is no longer available. We use strategic link scores to (i) explain black-box RL agents by identifying strategically-linked pairs among decisions they make, and (ii) improve the worst-case performance of decision support systems by distinguishing whether recommended actions can be adopted as standalone improvements, or whether they are strategically linked hence require a commitment to a broader strategy to be effective. We demonstrate these use cases with maze-solving and chess-playing examples as well as simulated healthcare and traffic environments.

## 1 INTRODUCTION

Being able to understand an RL policy is essential in many high-stakes environments. For example, in healthcare, the responsibility for making good treatment decisions rests with the clinician. Thus, the clinician will want to understand the policy prior to adopting any of its recommendations.

What, however, does it mean to understand a policy? In supervised learning, each individual model output is informed just by its corresponding input. However, policies are coordinated actions optimized to achieve long-term goals. Even if the policy function was inherently interpretable, it would still be non-obvious what actions are meant to work together to achieve a goal.

In this work, we address this unique challenge by developing a method to expose the strategic links between the actions of a decision-making policy. Specifically,

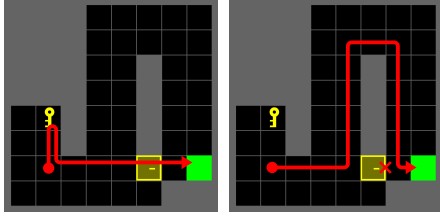

(a) **Optimal Strategy**  (b) **Shortcut Blocked**

Figure 1: *Strategy in a navigation task.* Picking up the key takes extra time early on but unlocks a major shortcut later. If the shortcut were to be blocked, going for the key would no longer be optimal. This shows that picking up the key and taking the shortcut are strategically linked—the key is picked up to take the shortcut.

we observe that early actions are often taken at the expense of short-term benefit under the *assumption* that they will enable future actions with greater overall returns. As a concrete example, consider a simple navigation task (Figure 1), where the straightforward path to the objective is long and winding. Slightly off that path, however, there is a key that unlocks a significant shortcut. In this scenario, the plan to retrieve the key—incurring a small delay—and then take the shortcut—saving more time than lost in the delay—is a strategic one. How do we know these actions are linked? If the shortcut were to be blocked, then an optimal agent would no longer choose to retrieve the key.

Let us refer to "retrieving the key" as the *set-up decision* and "taking the shortcut" as the *pay-off decision*. **This brings us to our first contribution, the formalizing of *strategically linked* actions:**

> The *strategic link score* between a set-up decision $a$ and a pay-off decision $a'$ is the *the drop in the likelihood of the set-up decision under the constraint that the pay-off decision is no longer available.*

This drop would be large when the set-up decision is not advantageous on its own but rather made primarily to facilitate the pay-off decision. In contrast, if the optimality of the two decisions is not contingent on each other— that is, the "set-up" decision is advantageous regardless of whether it is followed up by the "pay-off" decision—then the drop would be zero.

Having a quantitative definition of strategic links empowers several additional contributions, including *planning-level explanations* of decision-making policies as well as provide *strategy-aware decision support* in high-stakes environments:

- **Planning-Level Explanations.** Existing methods for explainable RL typically provide either state-level explanations—such as important features of a state that most influence the corresponding action (e.g. (Greydanus et al., 2018))—or policy-level explanations—such as key trajectories that summarize a policy's behavior (e.g. (Amir and Amir, 2018)). In contrast, our strategic link scores provide *planning-level explanations* that allow the user to understand if and when some recommended action now is conditional on some other action being available in future.

- **Strategy-Aware Decision Support.** In high-stakes environments, it is rarely feasible to deploy a fully autonomous RL agent. More common are decision support systems, where the RL policy recommends actions to a human decision-maker, who then judges what final decision to make (e.g. Jones et al., 2023). However, suppose that the recommended action now was to set up for another action later, but the user unknowingly only implements the first action and not the second—and ends up worse off as a result. Our strategic link scores can be used to inform users when sets of actions must be implemented together to achieve a beneficial outcome and when implementation decisions can be made independently. Through experiments, including in a healthcare environment based on the MIMIC-IV dataset (Johnson et al., 2023), we show that providing strategy-aware recommendations leads to greater improvements in the original decision-maker's performance, both on average and in worst-case scenarios.

Finally, while the two use cases we highlight are geared towards practical use cases of experts trying to productively work with RL agents, our definition of strategic link scores can be used to understand the outputs of any type of planner, regardless of whether it is optimal or even reward-based. For example, in Section 6, we use our strategic link scores to expose the planning horizon associated with the emergent traffic routing policy that arises from the many individual routing decisions made by cars in a realistic traffic simulator. This example gives a taste of the many analyses that our strategic link scores may enable beyond planning-level explanations and strategy-aware decision support.

## 2 RELATED WORK

Our work is related to *explainable RL* (at state, policy, planning levels) and *safe policy improvement*.

**State-Level XRL.** State-level XRL explain *individual* decisions: why a particular state is mapped to its corresponding action. A common approach is to adapt techniques from supervised settings, such as extracting saliency maps (e.g. Greydanus et al., 2018; Iyer et al., 2018), collecting human annotations (e.g. Ehsan et al., 2018), fitting white-box meta-models (e.g. Liu et al., 2018), or generating counterfactual examples (e.g. Olson et al., 2021; Chen et al., 2022; Huber et al., 2023). While this last group of works consider *counterfactual states*, strategic links involve *counterfactual policies* that could have been optimal had the environment conditions been different.

**Policy-Level XRL.** These explanations consider *all* the decisions of a given policy collectively. Some methods summarize those decisions via key trajectories (Amir and Amir, 2018) or policy graphs (Topin and Veloso, 2019); others contrast the policy with a baseline (van der Waa et al., 2018; Yao et al., 2022); and many aim to learn white-box policies directly (e.g. Khan et al., 2009; Shu et al., 2018; Verma et al., 2018; Hein et al., 2018; Silva et al., 2020; Sun et al., 2023). All of these methods are limited in scope to a single policy and ignore the planning process behind it. In contrast, planning-level explanations such as strategic links consider policies in context of others related through a shared planning process.

**Planning-Level XRL.** Closer to the spirit of our work, several methods explain RL policies by describing how a particular action choice impacts the future rewards an agent will attain, whether broken down by reward dimension (Juozapaitis et al., 2019; Erwig et al., 2018) or simply on the probability of success (Madumal et al., 2020; Yau et al., 2020; Cruz et al., 2023). In contrast, our strategic links expose how a particular action choice sets up for future actions—enabling a

fundamentally different kind of policy understanding. Moreover, our strategic link scores can be computed for any type of planner, as long as their planning process can be intervened on, even if it does not contain an explicit reward.

**Safe Policy Improvement.** Improving upon policies already in deployment is a common use case of RL. A key challenge in this setting is safety: ensuring the new policy does not perform worse than the original when deployed. This is usually a risk caused by the poor coverage of the available data collected by the original policy. So, prior work tends to focus on solutions at the training stage—for instance, limiting deviations from the original policy to keep new policies within coverage region (Laroche et al., 2019; Wu et al., 2022; Sharma et al., 2024). Strategy-aware recommendations, on the other hand, address the post-training risk of improper or partial adoption of new policies by flagging changes that are strategically linked and should be implemented together.

## 3 PRELIMINARIES

**Environments.** We consider a decision-making environment $\mathcal{E} = (\mathcal{S}, \mathcal{A}, \sigma, \tau)$, where $\mathcal{S}$ is the *state* space, $\mathcal{A}$ is the *action* space, $\mathcal{A}(s) \subseteq \mathcal{A}$ denotes the set of actions available in state $s \in \mathcal{S}$, $\sigma \in \Delta(\mathcal{S})$ is the *initial state distribution*, and $\tau : \mathcal{S} \times \mathcal{A} \to \Delta(\mathcal{S})$ is the *transition function*. A decision-maker interacts with this environment over time. Initially, the environment is in state $s_1 \sim \sigma$. At each time step $t \in \{1, 2, \ldots\}$, the decision-maker observes the current state $s_t$, takes one of the available actions $a_t \in \mathcal{A}(s_t)$, and the environment transitions into a new state $s_{t+1} \sim \tau(s_t, a_t)$.

**Policies & Planners.** During this interaction, the decision-maker chooses actions according to a *policy* $\pi$, which is produced by a *planning process (planner)* $\mathcal{P}$. Policies $\pi : \mathcal{S} \to \Delta(A)$ map each state to a distribution over actions such that $a_t \sim \pi(s_t)$, and we denote with $\pi(a|s)$ the probability of action $a$ being taken in state $s$. Meanwhile, planners $\mathcal{P}$ are processes that output policies given an environment $\mathcal{E}$, and we denote with $\pi = \mathcal{P}(\mathcal{E})$ the policy planned by $\mathcal{P}$ for environment $\mathcal{E}$. For example, reinforcement learning is concerned with optimal planning—the *optimal planner* $\mathcal{P}_r^*$ aims to find policies that maximize the expected *value* $v$ under some *reward function* $r : \mathcal{S} \times \mathcal{A} \to \mathbb{R}$:

$$\pi_r^* = \mathcal{P}_r^*(\mathcal{E}) = \operatorname{argmax}_\pi \mathbb{E}_{\pi,\mathcal{E}}\big[\, v \doteq \textstyle\sum_t r(s_t, a_t) \,\big] \tag{1}$$

Although the optimal planner serves as a useful example, it is important to also keep in mind that not all planners are optimal and some may not even seek to maximize a reward function.

**Constrained Actions.** Our characterization of strategies involve hypothetical policies that would have been planned if certain actions had been unavailable to the planner. To capture this, we define $\mathcal{E}_{\neg\tilde{a}|\tilde{s}}$ as a constrained version of $\mathcal{E}$ with action $\tilde{a}$ removed from the available actions $\mathcal{A}(\tilde{s})$ in state $\tilde{s}$:

$$\mathcal{E}_{\neg\tilde{a}|\tilde{s}} = (\mathcal{S}, \mathcal{A}[\tilde{s} \mapsto \mathcal{A}(\tilde{s}) \setminus \{\tilde{a}\}], \sigma, \tau) \tag{2}$$

where $\mathcal{A}[\tilde{s} \mapsto \mathcal{A}(\tilde{s}) \setminus \{\tilde{a}\}](s)$ is equal to $\mathcal{A}(\tilde{s}) \setminus \{\tilde{a}\}$ for $s = \tilde{s}$ but equal to $\mathcal{A}(s)$ for all $s \in \mathcal{S} \setminus \{\tilde{s}\}$. Then, $\pi_{\neg\tilde{a}|\tilde{s}} = \mathcal{P}(\mathcal{E}_{\neg\tilde{a}|\tilde{s}})$ becomes the policy that would have been planned by $\mathcal{P}$ if action $\tilde{a}$ were to be unavailable in state $\tilde{s}$ in the original environment $\mathcal{E}$. Following our previous example, for the optimal planner $\mathcal{P}_r^*$, this would be the policy optimized with the constraint that $\pi(\tilde{a}|\tilde{s}) = 0$:

$$\pi_{\neg\tilde{a}|\tilde{s}}^* = \mathcal{P}_r^*(\mathcal{E}_{\neg\tilde{a}|\tilde{s}}) = \operatorname{argmax}_\pi \mathbb{E}_{\pi,\mathcal{E}_{\neg\tilde{a}|\tilde{s}}}[v] = \operatorname{argmax}_{\pi:\pi(\tilde{a}|\tilde{s})=0} \mathbb{E}_{\pi,\mathcal{E}}[v] \tag{3}$$

## 4 STRATEGIC LINK SCORES

We define a *decision* $(s, a)$ as the action $a$ taken in state $s$. For a given planner $\mathcal{P}$ and environment $\mathcal{E}$, we define the *strategic link score* between *set-up decision* $(s, a)$ and *pay-off decision* $(\tilde{s}, \tilde{a})$ as

$$\mathfrak{S}_{s,a \to \tilde{s},\tilde{a}} = \pi(a|s) - \pi_{\neg\tilde{a}|\tilde{s}}(a|s) \quad where \quad \pi = \mathcal{P}(\mathcal{E}), \; \pi_{\neg\tilde{a}|\tilde{s}} = \mathcal{P}(\mathcal{E}_{\neg\tilde{a}|\tilde{s}}) \tag{4}$$

This measures the drop in the likelihood of the set-up decision, $\pi(a|s)$, caused by constraining the pay-off decision in $\mathcal{E}_{\neg\tilde{a}|\tilde{s}}$. A large drop implies that the set-up is pursued primarily when it can be followed up by the pay-off, indicating a strong strategic link between the two decisions.

Crucially, the strategic link score $\mathfrak{S}$ is a property of the planner $\mathcal{P}$ and not solely of the environment $\mathcal{E}$. This means two planners performing different tasks in the same environment—or even the same task, at different levels of optimality—can have different strategic links between their decisions. Moreover, being a property of the planner, policy $\pi$ alone does not provide enough information to determine strategic links. We also need to know how that policy would change if certain decisions were constrained (as in $\pi_{\neg\tilde{a}|\tilde{s}}$).

**Illustrative Example.** Interestingly, two planners might be following the exact same policy and still have different strategic links depending on how they respond to the same constraint. Consider an environment with two states $\{1^{st}, 2^{nd}\}$ and two actions $\{\Omega, \Lambda\}$. The initial state is always $1^{st}$, and the transitions are deterministic (see Figure 2). We are given two different reward functions, $r$ and $\rho$, as in Figure 2. Suppose two optimal planners, $\mathcal{P}_r^*$ and $\mathcal{P}_\rho^*$, maximize $r$ and $\rho$ respectively over a two-step horizon. Under either reward function, the optimal policy is exactly the same: take action $\Lambda$ in $s = 1^{st}$ and then action $\Lambda$ again in $s = 2^{nd}$. Letting $\pi^r = \mathcal{P}_r^*(\mathcal{E})$ and $\pi^\rho = \mathcal{P}_\rho^*(\mathcal{E})$,

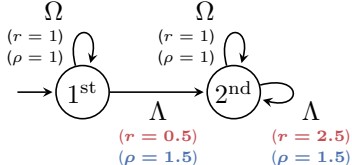

Figure 2: *Dynamics of the illustrative example. Starting in $s_1 = 1^{st}$, the action sequence $(\Omega, \Omega)$ is optimal under both reward functions $r$ and $\rho$. For $r$, the two actions are strategically linked as $\Lambda|1^{st}$ is not optimal unless $\Lambda|2^{nd}$. For $\rho$, there is no strategic link as $\Lambda|1^{st}$ is optimal regardless of the action taken in $s = 2^{nd}$.*

$$\pi^r(1^{st}) = \pi^\rho(1^{st}) = \Lambda; \quad \pi^r(2^{nd}) = \pi^\rho(2^{nd}) = \Lambda \quad (5)$$

Despite having the same policies, for planner $\mathcal{P}_r^*$, these two decisions are strategically linked. Under $r$, the immediate reward of $\Omega$ is larger than $\Lambda$ in $s = 1^{st}$: $r(1^{st}, \Omega) = 1 > r(1^{st}, \Lambda) = 0.5$. However, $\mathcal{P}_r^*$ still takes action $\Lambda$ to be able to transition into $s = 2^{nd}$, where an even larger pay-off is available: $r(2^{st}, \Lambda) = 2.5$. If taking action $\Lambda$ in $s = 2^{nd}$ had not been an option, taking action $\Omega$ in $s = 1^{st}$ would no longer have been optimal, as reflected in the strategic link score between the two decisions:

$$\mathfrak{S}_{(1^{st}, \Lambda) \to (2^{nd}, \Lambda)}^r = \underbrace{\pi^r(\Lambda|1^{st})}_{= 1} - \underbrace{\pi_{\neg(\Lambda|2^{nd})}^r(\Lambda|1^{st})}_{= 0} = 1 \quad (6)$$

But for planner $\mathcal{P}_\rho^*$, there is no strategic link. Under $\rho$, transitioning into $s = 2^{nd}$ is not important as the rewards stay the same in either state: $\rho(1^{st}, \Omega) = \rho(2^{nd}, \Omega) = 1$ and $\rho(1^{st}, \Lambda) = \rho(2^{nd}, \Lambda) = 1.5$. Taking action $\Lambda$ in $s = 1^{st}$ is optimal regardless of what action is taken in the next state $s = 2^{nd}$ and

$$\mathfrak{S}_{(1^{st}, \Lambda) \to (2^{nd}, \Lambda)}^\rho = \underbrace{\pi^\rho(\Lambda|1^{st})}_{= 1} - \underbrace{\pi_{\neg(\Lambda|2^{nd})}^\rho(\Lambda|1^{st})}_{= 1} = 0 \quad (7)$$

# 5 PRACTICAL APPLICATIONS

Having defined the strategic link score $\mathfrak{S}$, we now illustrate two of its applications: planning-level explanations (Section 5.1) and strategy-aware decision support (Section 5.2). We present the main results here and provide detailed setup information in the appendix. For the time being, we assume the planner $\mathcal{P}$ to be known (an RL algorithm with known reward functions). In the subsequent section, we will discuss how to compute strategic link scores for unknown planners as well.

## 5.1 PLANNING-LEVEL EXPLANATIONS

We first use strategic link scores to find pairs of set-up and pay-off decisions. These pairs expose how the actions of a planner are interlinked with each other, rather than treating actions in isolation (state-level explanations) or merely showing actions that occur together (policy-level explanations).

**Abstract Environment: GridWorld.** In `GridWorld`, the agent incurs a constant negative reward per timestep until it reaches the target. The agent can move up, down, left, or right, and by first walking over keys, they can later pass through previously locked doors. Our goal is to explain policies learned by soft value iteration (Haarnoja et al., 2017) for various maze layouts. We make sure none of the layouts become unsolvable if a particular decision is constrained.

**Realistic Environment: Chess.** In `Chess`, two players, white and black, take turns making moves. They both evaluate all board states after each potential move and soft-max the board values to obtain a distribution over next moves. This procedure constitutes the planner.

---
**Algorithm 1**
Planning-Level Explanations

1: **Input:** Planner $\mathcal{P}$, environment $\mathcal{E}$
2: $s_1 \sim \sigma_1$
3: **for** $t \in \{1, \dots, T\}$ **do**
4: $\quad a_t \leftarrow \operatorname{argmax}_{a \in \mathcal{A}} \pi(a|s_t)$
5: $\quad s_{t+1} \sim \tau(s_t, a_t)$
6: **for** $t \in \{1, \dots, T\}$ **do**
7: $\quad$ **for** $t' \in \{t, \dots, T\}$ **do**
8: $\quad\quad \tilde{\mathfrak{S}}_{tt'} \leftarrow \mathfrak{S}_{(s_t, a_t) \to (s_{t'}, a_{t'})}$
9: **Output:** Scores $\tilde{\mathfrak{S}}$

---

**Method.** Given a planner $\mathcal{P}$, we explain its policy $\pi = \mathcal{P}(\mathcal{E})$ by first selecting a trajectory representative of its behavior, and then computing the strategic link score between each pair of decisions along that trajectory. We consider trajectories, rather than random decisions, as they are more likely to exhibit strategic links. While any trajectory of interest can be analyzed in the same way, for this

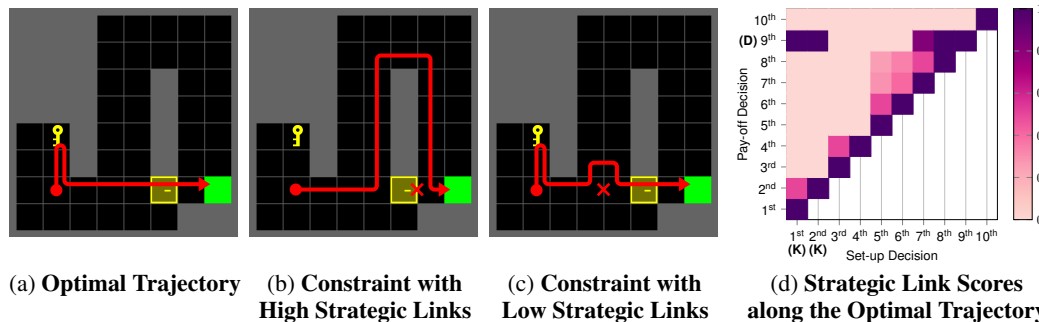

(a) **Optimal Trajectory** (b) **Constraint with High Strategic Links** (c) **Constraint with Low Strategic Links** (d) **Strategic Link Scores along the Optimal Trajectory**

Figure 3: *Strategic link scores for a simple maze layout.* Creating a shortcut by picking up the key to unlocking the door is strategic (a), since blocking the door results in the key not being picked up (b), while constraining an unrelated action does not lead to the same outcome (c). By looking at the strategic link scores between all the decisions along the optimal trajectory (d), the link between the key ("K") and the door ("D") can be seen clearly.

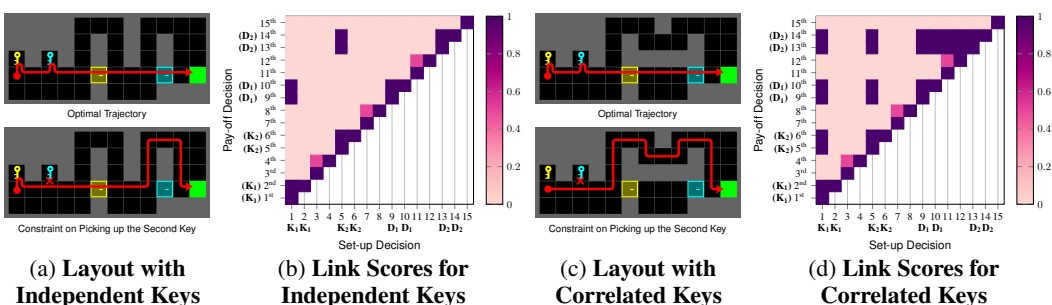

(a) **Layout with Independent Keys** (b) **Link Scores for Independent Keys** (c) **Layout with Correlated Keys** (d) **Link Scores for Correlated Keys**

Figure 4: *Strategic link scores for layouts with independent vs. correlated keys.* When the keys unlock separate shortcuts (a), they is no strategically link between them (b). If one key is skipped, collecting the other still remains optimal. When the keys jointly unlock a single shortcut (c), the decisions to pick up each key are strategically linked (d). If one key is skipped, collecting the other becomes pointless.

demonstration, we use the most likely trajectory, obtained by taking the highest-probability action at each step. When computing scores, we treat the earlier decision in each pair as the set-up and the latter as the pay-off. The complete procedure is given in Algorithm 1.

**In GridWorld, strategic link scores reflect the set-up and pay-off link between keys and doors.**
Figure 3 shows our approach in action for the maze layout from our introduction. In Figure 3a, we plot the most likely trajectory under the soft optimal policy: The target can be reached via a long and winding path, but a quicker strategy is to pick up a key that is slightly out of the way to unlock a door, creating a significant shortcut. We know the key and the door are strategically linked because if we block the door, picking up the key is no longer optimal, see Figure 3b. Preventing another decision that is not related to the key does not lead to the same outcome, see Figure 3c. In Figure 3d, we plot the strategic link scores between all potential set-up and pay-off decisions along the optimal trajectory, which makes it clear that moving towards the key (1st and 2nd decisions, labeled "K") are indeed a set-up for going through the door (9th decision, labeled "D").

**In GridWorld, strategic link scores can disambiguate whether two actions are linked or not.**
In Figure 4, we consider two other layouts, each with an additional key and its corresponding door. In the first layout (Figure 4a), the two keys unlock separate shortcuts, so whether one of them is retrieved or not has no effect on the optimality of retrieving the other. In Figure 4b, we see that each key ("$K_1$" and "$K_2$") is strategically linked to its corresponding door ("$D_1$" and "$D_2$"), but not to the other key-door pair. In the second layout (Figure 4a), there is just a single shortcut and unlocking it requires both keys to be collected. Now, the keys are strategically linked: If one of them is not retrieved, there is no point in retrieving the other (seen in Figure 4d). These two layouts also highlight the importance of strategy-aware recommendations. If an agent is not taking advantage of any shortcuts by collecting keys, we might recommend them to pick up both keys. In the independent case, it is safe for the agent to only follow through on just collecting one of the recommended keys. In the strategically linked case, however, if the agent were to collect only one key without the other, they would unnecessarily waste time as the shortcut would remain inaccessible. We will see the benefit of this distinction when providing decision support in Section 5.2.

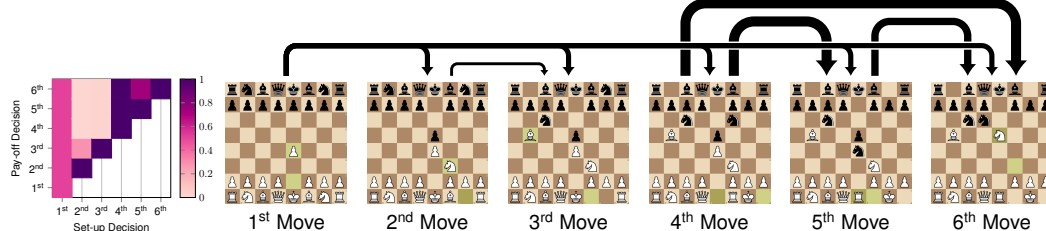

Figure 5: *Strategic Links between White Moves during a Chess Opening.* Arrows represent set-up and pay-off relationship, the boldness being determined by the strength of the strategic link. White employs one notable strategy: bringing their rook closer to the center by castling (4th move), threatening the black knight with that rook (5th move), and supporting with the same rook the white knight's capture of the black pawn (6th move).

**In Chess, identifying strategies expose what actions are planned together.** Allowing each side to make six moves resulted in the opening sequence in Figure 5, which also shows the strategic link scores between these moves from both white's and black's perspectives. White employs one notable strategy: In their 4th to 6th moves, they first castle to bring their rook closer to the center files, then threaten the black knight with their better positioned rook. The black knight flees, allowing the white knight to capture a pawn with the support the white rook (which prevents the black knight from capturing back). We can make this interpretation because

## 5.2 APPLICATION: STRATEGY-AWARE DECISION SUPPORT

In this section, we consider decision support systems and use strategic link scores to highlight when recommended actions are strategically linked to each other. This help avoid cases where actions recommended as set-ups are implemented without their pay-offs.

**Abstract Environment: Shortcuts.** We consider a procedurally-generated collection of abstract environments that are designed to have dynamics similar to key-door relationships from our previous application, called Shortcuts. (see example in Figure 6). The agent starts at some initial node and needs to move one node at a time until they reach node $N$. Each move costs one unit of time but reaching the target returns a reward of $N$. What makes the environment interesting is there are shortcuts that jump over multiple nodes at once. A shortcut becomes

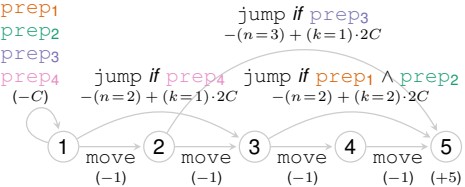

Figure 6: *An example environment that has 5 nodes, 3 shortcuts, and 4 preparation actions.* After taking the required preparation actions, the agent can jump via shortcuts, reaching the end more efficiently.

available after performing certain preparation actions (similar to keys in GridWorld). Preparation actions cost a fixed amount $C < 1/2$. They become advantageous through shortcuts they enable: Jumping forward via a shortcut that spans $n$ nodes and requires $k$ preparations costs $n - k \cdot 2C$ rather than $n$ individual moves. We generate 100 environments with 10 nodes, 5 shortcuts, and 5 preparation actions, setting $C = 0.1$. Which nodes the shortcuts jump over and which preparations they require are randomized, while preparation actions are always placed at the initial node.

**Realistic Environment: Hypotension.** We consider a healthcare scenario, called Hypotension, about assigning treatments to ICU patients with hypotension, which is based on a realistic simulator created by fitting a Markov decision process (MDP) to the *real* health record data from the MIMIC-IV dataset. As the state space, we consider discretized measurements of four biomarkers: *partial pressure of $O_2$ over fraction of inspired $O_2$*, *mean blood pressure*, *Glasgow Coma Scale*, *creatinine levels* ($|\mathcal{S}| = 36$). As the action space, we consider two binary treatments: administering *vasopressor therapy* or *intravenous fluid bolus* ($|\mathcal{A}| = 4$). The reward function is a weighted combination of the four measurements that constitute the state space, pushing each biomarker towards what is considered healthy for that biomarker. A description of the exact MDP can be found in the appendix.

**Methods.** Our goal is to recommend actions to a suboptimal agent to improve their performance. For Shortcuts, we consider an agent that knows of shortcuts and makes use of them when they are available, however, their behavior suboptimal because they are not aware of the required preparations—that is, they never take a preparation action unless one is recommended by us. Hence, a recommendation becomes a set of specific preparation actions. For instance, in Figure 6, the two shorter jumps happen to be more efficient, so the optimal recommendations are $\{\texttt{prep}_1, \texttt{prep}_2, \texttt{prep}_4\}$.

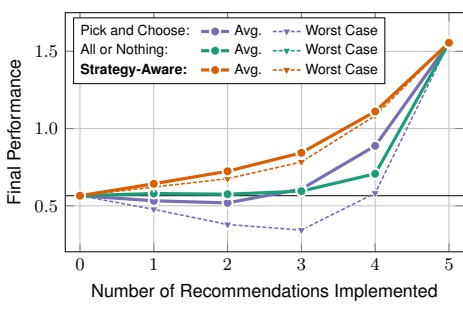 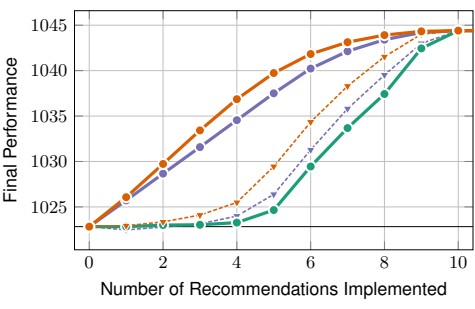

(a) **Results for Shortcuts**          (b) **Results for Hypotension**

Figure 7: *Performance following various recommendation methods.* In (a) `Shortcuts`, Pick-and-Choose is not safe, potentially leading to worse performance than to begin with. All-or-Nothing is safe but not effective unless a large number of recommendations are implemented. Strategic-Aware is both safe and effective for smaller number of implemented recommendations. In (b) `Hypotension`, Pick-and-Choose happens to be safe but Strategy-Aware is still an improvement (especially in worst-case scenarios for low adoption rates).

For `Hypotension`, we create 100 suboptimal agents by adding Gaussian noise to the true reward function and training RL agents using the resulting noisy reward functions. Whenever, in a state, a suboptimal agent's most likely action differs from the optimal action in a state, we recommend them to take the optimal action in that state. So, our recommendations form a list of state-action pairs.

Given a set of policy change recommendations, we consider three ways to share them with the user:

(i) **Pick-and-Choose**: The user can pick which changes to implement. The risk is if two changes are strategically linked, the user might implement only one and end up worse off. For evaluation, we consider all possible combinations and report average as well as worst-case performances.

(ii) **All-or-Nothing**: One can inform the user that the recommendations might be related, and thus must be implemented all together or not at all. This can be overly cautious: If the user does not want to make many changes, they might miss out on individual improvements that are not reliant on others.

---

**Algorithm 2**
Strategy-Aware Recommendations

1: **Input:** Planner $\mathcal{P}$, its recommended decisions $\mathcal{D} = \{(s,a)_i\}_{i=1}^N$
2: **for** $i \in \{1, \ldots, N\}$ **do**
3:     $\mathcal{D}_i \leftarrow \{(s,a)_i\}$
4:     **for** $j \in \{1, \ldots, N\}$ **do**
5:         **if** $\mathfrak{S}_{(s,a)_i \to (s,a)_j} \gg 0$ **then**
6:             $\mathcal{D}_i \leftarrow \mathcal{D}_i \cup \{(s,a)_j\}$
7: **Output:** $\{\mathcal{D}_1, \ldots, \mathcal{D}_N\}$

---

(iii) **Strategy-Aware**: Each recommendation is considered as a potential set-up decision, and it is grouped together with every other recommendation that is strategically linked to it as a pay-off decision (see Algorithm 2). As an example, consider the recommendations for the environment in Figure 6. *Strategy-Aware* would group them as { {prep$_1$, prep$_2$}, {prep$_4$} }, while *Pick-and-Choose* is equivalent to the grouping { {prep$_1$}, {prep$_2$}, {prep$_4$} }, and *All-or-Nothing* is equivalent to { {prep$_1$, prep$_2$, prep$_4$} }.

**In Shortcuts, strategy-aware decision support avoids harm due to partial adoption of linked recommendations.** For each strategy, we consider all subsets of recommendations the agent can implement and average the mean and the worst-case performance across the 100 randomly generated environments. Figure 7a shows the results broken down with respect to the number of individual recommendations implemented. The worst-case performance of *Pick-and-Choose* tends to be worse than that of the original policy. *All-or-Nothing* is safer, never resulting in worse performance, as it does not allow for partial adoption of recommendations. However, it leads to significant improvements only if a large number of recommendations are implemented. Meanwhile, our *Strategy-Aware* approach, which takes advantage of our concept of strategically-linked actions, is safe and effective even when the user only wishes to make a few changes to their policy.

**In Hypotension, strategy-aware decisions support is beneficial even when partial adoptions are not actively harmful.** In Figure 7b, we plot the same quantities as earlier but for `Hypotension`, calculating means across the 100 suboptimal agents we have trained using noisy reward functions. As it happens, a partial adoption of recommendations is almost never harmful in `Hypotension`; the worst-case performance for *Pick-and-Choose* does not drop below the initial performance (except slightly when only a single recommendation is implemented). However, *Strategy-Aware* still leads to better improvements over the baseline performance (both on average and in the worst-case scenarios).

## 6 VERSATILITY OF STRATEGIC LINK SCORES ACROSS PLANNING SETTINGS

We now describe how our strategic link scores can be applied across a range of planning settings. Specifically, our approach to defining strategic link scores enables us to reason about planners even when the planning process is a potentially complex, reward-less black box (e.g. the output of a traffic simulator, Section 6.1). We also demonstrate that we can infer strategic link scores from a finite batch of trajectories from *one* policy via inverse reinforcement learning (IRL, Section 6.2).

### 6.1 STRATEGIC LINK SCORES FOR REWARDLESS, BLACKBOX PLANNER VIA INTERVENTIONS

Realistic traffic simulators, such as `UXsim` (Seo, 2025), have explicit rules for an individual driver's behavior. However, how these rules executed by many individual drivers collectively shape the traffic flow policy is difficult to track formally, resulting in an unknown planner. Below, we construct strategic link scores for this blackbox, rewardless planning setting, and demonstrate that we can infer interesting properties of the planner such as how far ahead drivers plan when making their routing decisions.

**Road Network.** We consider the road network in Figure 8, which we call `ArterialHighway`. Each driver can use either the arterial road (slower but shorter) or the highway (faster but longer) to get from the entry to the exit; along the arterial road, there are junctions, "J1" to "J10", where drivers can switch onto the highway. If there is no congestion, then the arterial road is the optimal route. However, if one knows that part of the arterial road is congested or clogged, then diverting to the highway as early as possible is optimal to take advantage of the higher speed limit.

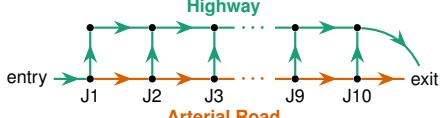

Figure 8: *The traffic scenario.* 'Entry' to 'exit', drivers need to decide whether to stay on an arterial road or divert to the highway at junctions 'J1' to 'J10'. Despite having a lower speed limit, the arterial route is shorter hence overall quicker. But, if one of its segments were to be closed off, it is better to divert to the highway the soonest to make use of its higher speed limit for longer.

**Intervention.** Taking advantage of the road structure, we can compute strategic link scores—and thus measure the planning horizon of drivers—through a simple intervention: closing off the arterial road at the last junction and then observing how early drivers tend to change their route. In formal terms, let $\pi^\dagger : \mathcal{S} = \{\text{J1}, \ldots, \text{J10}\} \to \mathcal{A} = [0, 1]$ be the collective policy that emerges out of individual drivers' decisions such that $\pi^\dagger(\text{JX})$ is the frequency with which drivers take the arterial road at junction JX. Then, we are interested in the following strategic link score, modified slightly to fit the continuous action setting: $\mathfrak{S}^\dagger_{\text{JX}\to\text{J10}} = \pi^\dagger(\text{JX}) - \pi^{\dagger:\{\pi(\text{J10})=0\}}(\text{JX})$.

**Simulation.** We run a simulation for 50k time steps, where the intervention to close off the arterial road at J10 is performed at time step 10k. Figure 9 shows our raw data: the number of vehicles over time that pass through each junction and stay on the arterial road; we see a clear change in behavior after the intervention. By taking the average slope of these count-over-time plots, we obtain traffic flow rates—number of vehicles per time step—at each junction, before and after the intervention, either staying on the arterial road or switching to the highway (Figure 10a). By normalizing the flow rates at each junction so that they add up to one, we obtain the pre- and post-intervention routing policies (Figure 10b). Taking the difference between the two policies gives us the strategic link scores we are after (Figure 10c).

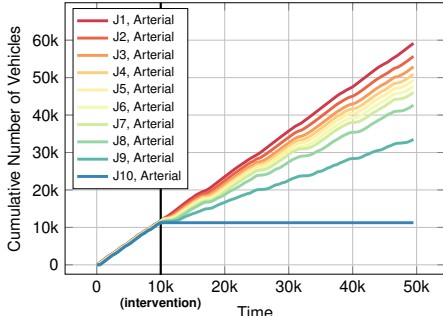

Figure 9: *Vehicle counts.* The rate at which vehicles pass through each junction and still stay on the arterial road changes significantly after the intervention at J10.

**Result: Blackbox, reward-free strategic link scores reveal planning horizon.** First, note the link score for J10 is one. This is by definition of link scores; every decision is strategically linked to itself with a score one. Since the intervention is performed at J10, every driver reaching J10 has no choice but to continue to the highway. Besides J10, we see that the most significant link score is at J9, corresponding to a 25% percentage-point drop in the rate of drivers that still decide to stay on the arterial road and continue to J10. This suggest that the collective behavior of drivers respond rather myopically to changes in traffic conditions. Compare their behavior to the optimal routing behavior in Figure 11. Besides J10, the only strategic link is to J1—the earliest diversion point—which requires a planning horizon of at least ten junctions.

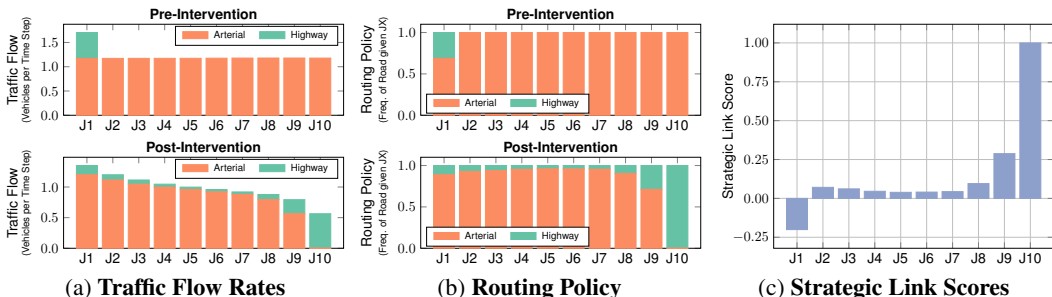

| (a) **Traffic Flow Rates** | (b) **Routing Policy** | (c) **Strategic Link Scores** |

Figure 10: *Strategic link scores for the simulated drivers.* Using the count data in Figure 9, we extract traffic flow rates (a), and normalizing those rates, we obtain the emergent routing policy of drivers, pre- and post-intervention (b). Strategic link scores are the difference between the two policies (c). When it comes to the decision of saying on the arterial road, the strongest strategic link to J10—besides J10 itself—is at J9. In other words, following the closure of the arterial road past J10, the drivers tend to divert to the highway mostly at J9.

While the emergent behavior of drivers is limited in horizon when diverting from the arterial road, one long-horizon effect still stands out: the significant *negative* link score at J1. This reflects drivers who were *avoiding* the arterial road even before the intervention, and whose *avoidance* was strategically linked to the road connection following J10. That connection is what made majority of the drivers prefer the arterial road, leading to congestion that slowed

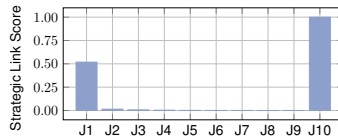

Figure 11: *Strategic link scores for optimal routing*—diverting at J1.

down the traffic enough for the longer highway route to become competitive. That is why some drivers decided to avoid the arterial road altogether. Once the intervention caused drivers to leave the arterial road, reducing congestion, this avoidance strategy became less frequent—hence the negative score.

## 6.2 STRATEGIC LINK SCORES FROM FINITE SAMPLES FROM A SINGLE POLICY VIA IRL

When a goal-driven planner is not available as an explicit algorithm but only observable through demonstrations—that is, trajectories generated by its unconstrained policy—we can still identify strategic links by first inferring a reward function that captures the planner's objective. We achieve this via maximum entorpy IRL (Ziebart et al., 2008). Because the coverage of demonstrations is critical for IRL, we experiment with different levels of stochasticity in actions by varying the temperature parameter in soft value iteration.

**Result: We can compute accurate strategic links from finite trajectories from one policy.** We evaluate inferred rewards using the EPIC distance, (Gleave et al., 2021), which accounts for equivalent reward expressions up to shaping, and measure the accuracy of strategic link scores computed based on those rewards using mean square error (MSE). In Figure 12, we see that demonstrations from *one* policy can carry enough information to model not just the policy of a decision-maker but also their planning process in the form of a reward function, and thereby identify strategic links.

## 7 CONCLUSION

We introduced *strategic link scores*, which formalize a key aspect of long-term planning: actions being taken to enable future actions. By identifying strategic links, we were able to provide planning-level explanations of RL agents, and even directly improve decision-making performance by guiding safe implementation of action recommendations. Beyond reward-based settings, we also used strategic link scores as a tool for characterizing planning processes that are not given as explicit algorithms.

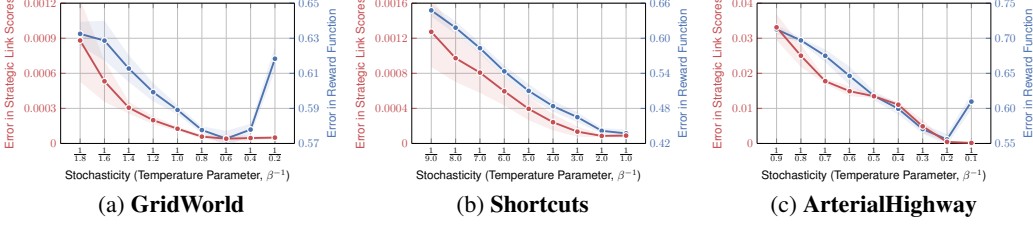

| (a) **GridWorld** | (b) **Shortcuts** | (c) **ArterialHighway** |

Figure 12: *Strategic link scores inferred from demonstrations* become more accurate with increasing variation, following a similar trend to reward inference (until policies are almost uniformly random and rewards become unidentifiable, strategic link scores remain accurate as recognizing policies to be uniformly random is sufficient).

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

## A EXPERIMENTAL DETAILS

### A.1 DETAILS FOR PLANNING-LEVEL EXPLANATIONS

**GridWorld.** The state space consists of the agent's position, represented as row and column indices $(i, j)$, and a binary flag $f_k$ for each key $k$, indicating whether the corresponding key has been retrieved. The action space consists of the four cardinal directions: up, down, left, and right. Transitions are deterministic. At each time step when the agent is not in the target cell, they receives a reward of $-1$. When collecting the most likely trajectory in Algorithm 1, we let the agent to interact with the environment until they reach the target, which determines the value of $T$.

As the planner, we consider soft value iteration (Haarnoja et al., 2017), setting the discount factor as $\gamma = 0.99$, the number of iterations as 250, and the inverse temperature parameter as $\beta = 100$. When computing strategic link scores, we constrain the decision $(s, a)$ by setting the reward value at that decision to negative infinity: $r(s, a) \leftarrow -\infty$. In GridWorld, we ignore flags $\{f_k\}$ and only consider the agent's position $(i, j)$ to be part of the state. That is, we consider moving a specific direction in a specific cell to be the same decision, regardless of which keys have been picked up so far. Accordingly, when constraining the decision $(i, j, \{f_k\}, a)$, we set $r(i, j, \{f'_k\}, a) \leftarrow -\infty$ for all $f'_k \in \{0, 1\}$, independent of the values $\{f_k\}$.

**Chess.** Players decide on moves by (i) considering the next board state after each legal move available to them, (ii) computing the value of those board states, and (iii) soft-maxing the computed values to obtain a distribution over which move to make next. When evaluating boards, we use a chess engine called Stockfish (https://github.com/official-stockfish/Stockfish), fixing its search depth to 20 and turning off threading and hashing options to get deterministic values back.

### A.2 DETAILS FOR STRATEGY-AWARE DECISION SUPPORT

**Shortcuts.** The state space consists of the node that the agent is currently at, along with binary flags corresponding to each preparation action, indicating whether that action has been taken or not. The action space consists of moving forward (move), jumping via the $i$-th shortcut ($\{\text{jump}_i\}$), and or taking the $j$-th preparation action ($\text{prep}_j$). Letting $n$ be the current node and $f_j$ be the flag corresponding to action $\text{prep}_j$, the transition dynamics are given by the following rules:

$$n \leftarrow n + 1 \quad \text{if} \quad a = \text{move} \ \wedge \mathcal{V}_{\text{move}} \doteq \{n \neq N\} \tag{8}$$

$$n \leftarrow n^{(\text{to})}_{\text{jump}_i} \quad \text{if} \quad a = \text{jump}_j \wedge \mathcal{V}_{\text{jump}_i} \doteq \{n = n^{(\text{from})}_{\text{jump}_i}\} \wedge \{f_j = 1, \forall j \in \mathcal{J}_{\text{jump}_i}\} \tag{9}$$

$$f_j \leftarrow 1 \quad \text{if} \quad a = \text{prep}_j \wedge \mathcal{V}_{\text{prep}_j} \doteq \{n = n_{\text{prep}_j}\} \tag{10}$$

where $\mathcal{V}_a$ denotes the validity condition for action $a$, $n^{(\text{to})}_{\text{jump}_i}$ is the destination of shortcut $i$, $n^{(\text{from})}_{\text{jump}_i}$ is the origin of shortcut $i$, $\mathcal{J}_{\text{jump}_i}$ is the set of preparation actions required to use shortcut $i$, and $n_{\text{prep}_j}$ is the node where action $\text{prep}_j$ is located at. Meanwhile, the reward function is given by

$$r(n, \{f_j\}, \text{move}) = \begin{cases} -1 & \text{if} \quad n + 1 \neq N \\ -1 + N & \text{if} \quad n + 1 = N \end{cases} \tag{11}$$

$$r(n, \{f_j\}, \text{jump}_i) = \begin{cases} -1 & \text{if} \quad \neg \mathcal{V}_{\text{jump}_i} \\ -(n^{(\text{to})}_{\text{jump}_i} - n^{(\text{from})}_{\text{jump}_i}) + |\mathcal{J}_{\text{jump}_i}| \cdot 2C & \text{if} \quad \mathcal{V}_{\text{jump}_i} \wedge n^{(\text{to})}_{\text{jump}_i} \neq N \\ -(n^{(\text{to})}_{\text{jump}_i} - n^{(\text{from})}_{\text{jump}_i}) + |\mathcal{J}_{\text{jump}_i}| \cdot 2C + N & \text{if} \quad \mathcal{V}_{\text{jump}_i} \wedge n^{(\text{to})}_{\text{jump}_i} = N \end{cases} \tag{12}$$

$$r(n, \{f_j\}, \text{prep}_j) = \begin{cases} -1 & \text{if} \quad \neg \mathcal{V}_{\text{prep}_j} \\ -C & \text{if} \quad \mathcal{V}_{\text{prep}_j} \end{cases} \tag{13}$$

We generate these environments randomly according to following procedure, letting $I$ be the number of shortcuts and $J$ be the number of preparation actions:

---

**Algorithm 3** `Shortcuts` Environment Generation

---

1: **for** $i \in \{1, \ldots, I\}$ **do**
2:     $n' \sim \text{Uniform}(\{1, \ldots, N\}); n'' \sim \text{Uniform}(\{1, \ldots, N\} \setminus \{n'\})$
3:     $n_{\text{jump}_i}^{(\text{from})} \leftarrow \min\{n', n''\}; n_{\text{jump}_i}^{(\text{to})} \leftarrow \max\{n', n''\}$
4:     $J' \sim \text{Uniform}(\{1, \ldots, J\}); \mathcal{J}_{\text{jump}_i} \leftarrow \{\}$
5:     **for** $_- \in \{1, \ldots, J'\}$ **do**
6:         $j' \sim \text{Uniform}(\{1, \ldots, J\} \setminus \mathcal{J}_{\text{jump}_i}); \mathcal{J}_{\text{jump}_i} \leftarrow \mathcal{J}_{\text{jump}_i} \cup \{j'\}$
7: **for** $j \in \{1, \ldots, J\}$ **do**
8:     $n_{\text{prep}_j} \leftarrow 1$

---

When using soft value iteration to plan policies, we set the discount factor as $\gamma = 0.99$, the number of iterations as 30 (twice the total number of nodes and preparation actions), and the inverse temperature as $\beta = 100$. When checking whether a strategic link score is significantly greater than zero, as in Line 5 of Algorithm 2, we set the significance threshold to halfway between $0$ and $1/5$ (one over the number of preparation action). This is because: When all preparation actions are recommended together, a soft-optimal policy assigns them equal probabilities at the initial state, since the order in which they are taken does not matter. Therefore, the strategic link score between any two preparation actions becomes at most $1/5$, as the probability of a recommendation cannot drop more than its original value after a constraint.

**Hypotension.** Starting with the MIMIC-IV dataset, we selected adults who (i) are aged 18 to 80, (ii) had an ICU stay that is at least 24 hours long, and (iii) exhibited mean arterial pressure (MAP) readings of 65mmHg or below, indicative of acute hypotension. After this filtering, we are left with 1684 distinct ICU admissions, each constituting a 'trajectory'.

We form states by discretizing four biomarkers, *partial pressure of $O_2$ over fraction of inspired $O_2$*, *mean blood pressure*, *Glasgow Coma Scale*, *creatinine levels*, according to the table below, resulting in tuples $s = (s[1], s[2], s[3], s[4]) \in \mathcal{S} = \{0, 1, 2\} \times \{0, 1\} \times \{0, 1\} \times \{0, 1, 2\}$.

Table 1: Discretization of Biomarkers

| Biomarker | Interval | Bin Value |
|---|---|---|
| Partial Pressure of $O_2$ / Fraction of Inspired $O_2$ | $[200, \infty)$ | $s[1] = 0$ |
| | $[100, 200)$ | $s[1] = 1$ |
| | $(-\infty, 100)$ | $s[1] = 2$ |
| Mean Blood Pressure | $[70\text{mmHg}, \infty)$ | $s[2] = 0$ |
| | $(-\infty, 70\text{mmHg})$ | $s[2] = 1$ |
| Glasgow Coma Scale | $(-\infty, 12]$ | $s[3] = 0$ |
| | $(12, \infty)$ | $s[3] = 1$ |
| Creatinine | $(-\infty, 1.9\text{mg/dL}]$ | $s[4] = 0$ |
| | $(1.9\text{mg/dL}, 4.9\text{mg/dL}]$ | $s[4] = 1$ |
| | $(4.9\text{mg/dL}, \infty)$ | $s[4] = 2$ |

The action space consist of two binary treatments, administering *vasopressor therapy* or *intravenous fluid bolus*, resulting in four distinct actions: administering no treatment, one or the other treatment, or both treatments. We infer the initial state distribution $\sigma$ and the transition function $\tau$ by counting initial states and transitions in the MIMIC-IV dataset. As the reward function, we consider $r(s) = 60 - 10s[1] - 10s[2] - 10s[4]$, which is inline with the fact that higher bin values signify a deterioration in the patient's condition.

As the optimal planner, we consider soft value iteration with discount factor $\gamma = 0.95$ and inverse temperature $\beta = 100$, running it for 250 iterations. We create suboptimal policies by adding Gaussian noise to the true reward function such that $r'(s) = r(s) + 2\eta_s$ and $\eta_s \sim \mathcal{N}(0, 1)$ and training "optimal" agents under the resulting noisy reward functions. We reject any suboptimal policy that differs from the optimal policy in more than 10 states.

## A.3 DETAILS FOR STRATEGIC LINK SCORES VIA INTERVENTIONS

**Simulation.** We refer to Seo (2025) for the details regarding the mechanisms underlying the traffic simulator. In our experiments, the simulation parameters are set as follows: `deltan = 5`, `reaction_time = 1`, `duo_update_time = 500`, `duo_update_weight = 0.5`, and `duo_noise = 0.01`. Each road segment along the highway, as well as the on-ramps leading to it, has two lanes, a length of 1000, and a free flow speed of 20. Road segments along the arterial have the same number of lanes and the same length, but a lower free flow speed of $20\sqrt{J/(J+1)}$. We choose this value so that the arterial road is reasonably quicker (takes less overall time): If the speed on the arterial road were to be $20 \cdot J/(J+1)$, both the arterial and highway routes would have taken the same amount of time as their length ratio is $J/(J+1)$. If it were to be 20 instead, so that the two routes have the same speed, shorter length would have always meant a quicker route, requiring no strategic routing. The actual free flow speed is the geometric mean of these extreme scenarios. Finally, the incoming traffic flow into the road network through the "entry" is set to be 2.0.

**Reinforcement Learning.** In addition to simulated agents, we also use RL to find the optimal routing policy that minimizes the average travel time throughout the whole road network. Each road segment has a travel time that depends on the traffic flow it carries, which we denote as $T_A(f)$ and $T_H(f)$ given flow $f$ for the arterial and the highway (including on-ramps) respectively. Since the incoming flow must be equal to the outgoing flow at each junction, the initial entry flow $f_0$ determines the flow throughout the network, conditioned on a routing policy $\pi$. Letting $f_{A,J_i}$, $f_{R,J_i}$, and $f_{H,J_i}$ denote the traffic flow on the arterial segment following junction $J_i$, the on-ramp leaving that junction, and the highway segment that follows, we can write

$$f_{A,J_1} = f_0 \cdot \pi(J_1) \qquad f_{R,J_1} = f_0 - f_{A,J_1} \qquad f_{H,J_1} = f_{R,J_1} \tag{14}$$
$$f_{A,J_i} = f_{A,J_{i-1}} \cdot \pi(J_i) \quad f_{R,J_i} = f_{A,J_{i-1}} - f_{A,J_i} \quad f_{H,J_i} = f_{H,J_{i-1}} + f_{R,J_i} \tag{15}$$
$$= \sum_{i'=1}^{i} f_{R,J_{i'}} = f_0 - f_{A,J_i} \tag{16}$$

Notice that the flow rates past each junction, $f_{A,J_i}$, $f_{R,J_i}$, $f_{H,J_i}$, can be determined solely from the flow rate coming into that junction, $f_{A,J_{i-1}}$. Leveraging this structure, we find the policy $\pi(J_i)$ that would result in the shortest average time efficiently via the following three steps: First, we include in our state space the incoming flow $f$ to junction $J_i$, for which we can write the following deterministic transition function:

$$\tau(J_i, f, a) = (J_{i+1}, f' = fa) \tag{17}$$

Second, we use value iteration to compute the optimal flow-dependent policy $\pi^*(J_i, f)$, which captures optimal routing under any flow condition, not just for our specific entry flow rate $f_0$. Third, we roll this policy out, starting from the entry flow rate $f_0$, to obtain a flow-free policy $\pi^*(J_i)$, which would remain optimal as long as the entry flow rate does not change:

$$\pi^*(J_1) \leftarrow \pi^*(J_1, f_0) \qquad f_{A,J_1} \leftarrow f_0 \cdot \pi^*(J_1) \tag{18}$$
$$\pi^*(J_i) \leftarrow \pi^*(J_i, f_{A,J_{i-1}}) \qquad f_{A,J_i} \leftarrow f_{A,J_{i-1}} \cdot \pi^*(J_i) \tag{19}$$

We use the following reward function

$$r(J_i, f, a) = \underbrace{fa \cdot T_A(fa)}_{\text{arterial}} + \underbrace{f(1-a) \cdot T_H(f(1-a))}_{\text{on-ramp}} + \underbrace{(f_0 - fa) \cdot T_H(f_0 - fa)}_{\text{highway}} \tag{20}$$

which ensures that cumulative rewards correspond to the average travel time across all road segments, weighted by the actual traffic flow carried by each segment. When using value iteration, we quantize the continuous actions space $\mathcal{A} = [0, 1]$ using 100 equally-spaced quantization points.

## A.4 DETAILS FOR STRATEGIC LINK SCORES VIA REWARD-BASED MODELING

For these experiments, we compute strategic link scores using reward functions inferred from demonstrations via IRL, rather than using the true reward functions. The specific scores we evaluate depends on the environment: For `GridWorld`, we consider the same score matrices as in Section 5.1; for `Shorcuts`, we consider the same scores that needed to be computed for the application in Section 5.2; and for `ArterialHighway`, we consider the same scores computed in Section 6.1.

Although compute the same scores as in our applications, we use slightly smaller versions of each environments to reduce the computational demand of performing IRL: For `GridWorld`, we consider only the simple layout shown in Figure 3; for `Shortcuts`, we generate 10 environments with 5 nodes, 3 shortcuts, and 3 preparation actions instead of 100 environments with 10 nodes, 5 shortcuts, and 5 preparation actions; for `ArterialHighway`, we consider a road network with 5 junctions instead of 10. In `ArterialHighway`, using soft value iteration requires us to quantize the continuous action space. We normally use 100 quantization points, but for these experiments, we reduce it to 10.

As demonstrations, we sample 10,000 trajectories following optimal policies computed via soft value iteration. In `GridWorld`, we fix the time horizon of each trajectory to $T = W \times H$, where $W$ is the grid width and $H$ is the grid height; in `Shortcuts`, we fix $T = N \times L$, where $N$ is the number of nodes and $L$ is the number of preparation actions; and in `ArterialHighway`, we fix $T = J + 1$, where $J$ is the number of junctions. For each environment, the number of soft value iteration steps is set accordingly. We vary the inverse temperature parameter to obtain demonstrations with different levels of stochasticity—see Figure 12 for the specific temperature values considered.

As the IRL algorithm, we use maximum entropy IRL (Ziebart et al., 2008). The RL algorithm used in the inner loop of maximum entropy IRLis again soft value iteration with the same hyperparameters (number of iterations and inverse temperature) as those used for generating demonstrations. Since all our environments have deterministic transitions, we consider the setting where the true transition dynamics are known. Meanwhile, reward functions are represented as $|\mathcal{S}|$-by-$|\mathcal{A}|$ matrices; initialized to all zeros; and updated via gradient steps over 10,000 iterations using the Adam optimizer with hyperparameters $\beta_1 = 0.9$, $\beta_2 = 0.999$, and $\epsilon = 10^{-8}$. The learning rate is set to $10^{-4}$ for `GridWorld` and `Shortcuts`, and to $5 \cdot 10^{-3}$ for `ArterialHighway`.

We repeat each experiment five times to obtain 1-sigma error bars.

