# OpenReview forum: "Strategically-Linked Decisions in Long-Term Planning and Reinforcement Learning"
_ICLR.cc/2026/Conference — Submitted to ICLR 2026_

### Official Review · Reviewer_MC5T · 2025-10-20

**Soundness:** 1
**Presentation:** 3
**Contribution:** 2
**Rating:** 2
**Confidence:** 4

**Summary:**

# Summary

The paper focuses on quantifying the causal dependencies between actions on a plan (policy).It formalizes the concept of strategically linked actions and defines strategic link score of two actions (producer and consumer).

It is worth noting that causal links were introduced in planning more than 30 years ago [1].
It seems like the authors have a very simplistic view of planning. While I understand that your setting does not assume state features, consider for a moment state features and actions having preconditions (in the simplest form, which of the state features are true) and effects (change in state features), as in e.g., classical planning. Then, an action would "produce" a feature that some other actions along the way would "consume" (require in the precondition). But it does not have to be pairs of actions, in the sense that a single action can achieve precondtions used by multiple actions along a plan (e.g., two doors opened by the same key), there can be alternative plans of the same cost that would use different actions that consume the same precondition (like two alternative doors). My point is, this property is not a property of action pairs. Looking at action preconditions and effects makes you see it more clearly. There was quite a body of research in planning on causal links, might worth a look.
Another closely related topic is action justification on a plan (e.g., [2,3,4,5,6]), talking about the reason for an action to be on a plan (providing some precondition for at least one following action or achieving a goal fact). The literature describes perfectly justified plans as a plan that cannot be reduced by removing actions and keeping it a plan. It seems like the authors are looking for similar concepts.


# Soundness

The main idea, strategic link score, as defined in Equation (4) does not make sense for planning in general, for the reasons described above. For instance, it is not hard to craft examples where the score will not drop when only one "pay-off" action is blocked. Of course, there are families of problems for which that is not possible, but I am not sure how to characterize such problems.

# Novelty

Novelty is limited, for the reason expressed above.

# Scholarship

[1] Weld, D. S. (1994). An Introduction to Least Commitment Planning. AI Magazine, 15(4), 27.
[2] Fink, E.; and Yang, Q. 1992. Formalizing Plan Justifications. In Proc. CSCSI 1992.
[3] Fink, E.; and Yang, Q. 1993.  A spectrum of plan justifications.  In Proceedings of the AAAI 1993 Spring Symposium,23–33.
[4] Lindner, F.; and Olz, C. 2022.  Step-by-Step Task Plan Explanations Beyond Causal Links. In31st IEEE InternationalConference on Robot and Human Interactive Communication, RO-MAN 2022
[5] Sreedharan, S.; Muise, C.; and Kambhampati, S. 2023. Generalizing Action Justification and Causal Links to Policies. In Proceedings of the Thirty-Third International Conference on Automated Planning and Scheduling (ICAPS).
[6] Salerno, M.; Fuentetaja, R.; and Seipp, J. 2023. Eliminating Redundant Actions from Plans using Classical Planning. In Proc. KR 2023, 774–778.

# Clarity

The presentation of ideas is sufficiently clear.

# Evaluation and Reproducibility

The evaluation is quite extensive, but meaningfulness of the results is questionable.

**Strengths:**

1. The authors present the ideas clearly
2. The paper include formal definitions
3. The paper presents a wide empirical evaluation

**Weaknesses:**

1. The related work ignores the literature on the same and related concepts that come from the field of planning
2. The authors present a very simplistic and unrealistic view of planning, as actions that are there to enable other actions in pairs
3. The main concept, "strategic link score" is not well-justified, as I mention above, as it is not a function of two actions
4. As a result of the former, the empirical evaluation does not provide meaningful insights

**Questions:**

1. Could you comment on the connection between your concept and the concepts from the planning literature above?
2. Could you comment on the soundness concerns?1. The related work ignores the literature on the same and related concepts that come from the field of planning
2. The authors present a very simplistic and unrealistic view of planning, as actions that are there to enable other actions in pairs
3. The main concept, "strategic link score" is not well-justified, as I mention above, as it is not a function of two actions
4. As a result of the former, the empirical evaluation does not provide meaningful insights

---

> ### Author Response · Authors · 2025-11-23
>
> **Novelty**
>
> Strategic link scores differ from existing work in classical planning in two important ways: (i) they relate actions to their necessary follow-up actions when it comes to achieving an objective, and (ii) they do not depend on any particular form of environment representation or planning procedure.
>
> Consider our toy example in Figure 2, where the decision sequence (1st, $\Lambda$) and (2nd, $\Lambda$) is optimal under both the red and blue reward structures. However, strategic link scores reveal that these actions work together under the red reward structure, whereas they are independent improvements over the alternative action $\Omega$ under the blue reward structure. Meanwhile, a precondition analysis would not differentiate these two cases. Under both reward structures, taking action $\Lambda$ in the 1st state “produces” the 2nd state, and taking action $\Lambda$ again “consumes” it. Performing (1st, $\Lambda$) is a precondition for (2nd, $\Lambda$) under either reward structure (as the transition dynamics stay the same). The novelty of strategic link decisions is capturing whether the two decisions (1st, $\Lambda$) and (2nd, $\Lambda$) are planned together, which is inherently a property of the planner rather than the environment.
>
> Is distinguishing whether two actions are planned together or not an important distinction to make? The answer is yes when recommending actions to an agent as the agent needs to know whether our recommendations must be implemented together to be effective or not. When we recommend Action A and Action B, it may be clear from the environment structure that Action A must be performed to be able to perform Action B (i.e. it is a precondition of Action B). What strategic link scores reveal is whether performing Action B is necessary for Action A to be a worthwhile action (or could the agent just take Action A without necessarily following it up with Action B).
>
> These relationships may seem trivial when the decision-making environment is specified formally as a set of features, with actions clearly producing and consuming these features. However, our definition is completely agnostic of what the environment is or how plans are formed. This is evident in our traffic example in Section 6.2, where we were able to reason solely from observable quantities that drivers using the arterial at J8/J9 relied on staying on the arterial road through J10.
>
> ---
>
> **Higher-Order Strategic Links**
>
> As the reviewer has pointed out, if an action A sets up either action B or C, constraints put just on B or just on C would not reveal the full relationship between A, B, and C.
>
> More generally, we formalized strategic links at the level of two individual state-actions but strategies can emerge at different levels of abstraction, and how to determine an appropriate level of abstraction remains an open question for future research. For example, in our maze environments, picking keys and using shortcuts were all associated with a single “move” action. But in a hierarchical RL setting, picking up a key could involve multiple sub-actions, such as a robot coordinating actuators to physically grasp the key. Then, the strategic link between picking a key and using a shortcut would be between these coordinated groups of sub-actions.
>
> That being said, our current definition provides an initial framework for formalizing strategic relationships. While looking at actions in pairs may seem simplistic, it still has practical uses as demonstrated by our applications, which vary from pedagogical settings like learning how to play chess, to making safe-yet-effective recommendations in healthcare, to characterizing how non-optimal agents make plans like drivers on the road.

---

> > ### Comment · Reviewer_MC5T · 2025-11-24
> > **I am not convinced (at least not yet) by your response**
> >
> > > Strategic link scores differ from existing work in classical planning in two important ways: (i) they relate actions to their necessary follow-up actions when it comes to achieving an objective, and (ii) they do not depend on any particular form of environment representation or planning procedure.
> >
> > Well, the point I made in my review (and you seem to agree) is that it is not a function of two actions, so (i) is not a valid point. Re (ii), this is also the case in classical planning, regardless of whether you *know* the preconditions and effects, they are there.
> >
> >
> >
> > > Consider our toy example in Figure 2
> >
> > This is actually precisely my point - the dependence of the second action's applicability on the first action's effects is not disappearing when you change the reward function, it is just merely *not recognized* by your definition. This remains true when you condition your definition on a policy.

---

> > > ### Author Response · Authors · 2025-12-02
> > >
> > > But the problem we want to solve is to come up with a metric that can disambiguate the two settings in Figure 2. For an optimal planner, when are actions greedily/individually optimal vs. only optimal when taken together? (More generally, when are the decisions of a planner independent from each other vs. tied to each other through the planning process?)
> > >
> > > Analyzing preconditions, while useful for other purposes, doesn't make this distinction. In both settings, the action in the first state is necessary to be able to take the same action in the second state, which is solely dictated by the transition function. However, depending on the planner and the reward function they are optimizing, it is either the case that (i) the optimal action in the first state is optimal regardless of what action is taken in the second state, or (ii) the optimal action in the first state is optimal only if the same action is also taken in the second state. Since this distinction is missed by existing approaches, including precondition analysis, developing a new metric such as strategic link scores is needed!

---

### Official Review · Reviewer_XeLC · 2025-10-28

**Soundness:** 3
**Presentation:** 4
**Contribution:** 2
**Rating:** 4
**Confidence:** 3

**Summary:**

The paper addresses the problem of explaining RL policies at the planning level, and proposes the strategic link score, which captures the temporal dependencies of the state-action pairs of a policy in terms of set-up and pay-off. The paper demonstrates how the strategic link score captures such dependencies in GridWorld and Chess environments. The paper also applies the proposed score for action recommendation, where the proposed score enables an appropriate level of granularity, forming a nice middle ground between recommending atomic actions and recommending full policies. The paper further demonstrates the utility of the proposed score for general non-reward-based planners, e.g., traffic simulators.

**Strengths:**

- The paper is visually appealing and very well-written, and key concepts are explained very clearly.
- Claims are well supported by evidence through well-explained demonstrations in a number of environments.
- The strategic link score could be of interest to researchers in explainable RL.

**Weaknesses:**

1. There seems to be a critical issue with the proposed method, which is that the strategic link score seems to fail whenever there exists more than one possible future pay-offs.
  - To give a concrete example, consider a GridWorld environment similar to that in Figure 3a. But now, instead of having a single door, there are two neighboring doors unlocked by the same key leading to the shortcut. Now, if we block one of the doors, the agent can still access the other door, so picking up the key is still part of the optimal strategy. Then, the probability of picking up the key is not decreased, and the strategic link score is 0, failing to connect the action of picking up keys to either of the future doors.
  - Then if we were to perform strategy-aware recommendation in this setting, it is possible for the agent to adopt the recommendation of picking up the key without adopting the actions of opening doors, causing a failure similar to pick-and-choose.
  - Whenever the future pay-off of a set-up can be realized in more than one way, the strategic link score fails to link the set-up to future pay-offs. This is because the score can only handle one-to-one correspondence of set-up and pay-off, and is incapable of linking a set-up to a combination of parallel/independent pay-offs.
2. The strategic link score seems to be limited to small tabular environments.
  - Assuming access to a planner (e.g., an RL algorithm) that returns a policy given an environment, to compute the strategic link score, you first use the the planner to return a policy, which is then used to generate a trajectory that is most likely under the returned policy. Then, for every state-action pair $(s, a)$ in the trajectory, you re-solve the environment while forbidding $(s, a)$ from being chosen by the policy. Then, the complexity of the method is $O(TP)$, where $T$ is the length of the trajectory, and $P$ is the complexity for the planner to solve the environment. In RL, for example, you would need to run an RL algorithm to solve MDPs $T$ times.
  - Such computation seems intractable for large MDPs that's costly to solve, and MDPs that have a long/infinite horizon.
  - Also, can the authors comment on the method's application on continuous state or action spaces? Is there a way to forbid an action in a continuous action space?
3. In relation to the above weakness, limitations is under-discussed in the paper.
4. Metrics of statistical significane (e.g., confidence intervals) are not reported in Figure 7.


Some minor issues

5. In the final paragraph on page 5, "In the second layout (Figure 4a)". Should be 4c instead of 4a.
6. On line 288, there's an unfinished sentence: "We can make this interpretation because"
7. Typo in the caption of Fig. 10: "When it comes to the decision of saying"    saying $\longrightarrow$ staying
8. First letter of "markov" should be capitalized in the references.

**Questions:**

1. In Figure 7, how is the baseline performance (black horizontal solid line) determined when you randomly generate the MDP? Is it the average of the performance of the 100 suboptimal agents?
2. For pick-and-choose and strategy-aware recommendation, how do you pick which of the recommended (group of) actions to incorporate when the number of recommendations implemented is smaller than the number of recommendations.
3. I do not follow how, in Figure 12, the error in strategic link score can further go down when the error in reward function increases as stochasticity becomes the highest.
4. The paper writes: "This is by definition of link scores; every decision is strategically linked to itself with a score one." However, by Eq. 4, isn't the link score of a state-action pair $(s, a)$ to itself $\pi(a | s) - 0 = \pi(a | s)$? When $\pi(a | s)$ was not 1 to begin with, how is the link score to itself 1?
5. What is the shaded region in Figure 12?

### Concluding Comments
Despite the paper being very well-written, due to the critical issue (see Weaknesses 1), I have provided an initial score of 4.

---

> ### Author Response · Authors · 2025-11-23
>
> **Higher-Order Strategic Links**
>
> As the reviewer has pointed out, if an action A sets up either action B or C, constraints put just on B or just on C would not reveal the full relationship between A, B, and C.
>
> More generally, we formalized strategic links at the level of two individual state-actions but strategies can emerge at different levels of abstraction, and how to determine an appropriate level of abstraction remains an open question for future research. For example, in our maze environments, picking keys and using shortcuts were all associated with a single “move” action. But in a hierarchical RL setting, picking up a key could involve multiple sub-actions, such as a robot coordinating actuators to physically grasp the key. Then, the strategic link between picking a key and using a shortcut would be between these coordinated groups of sub-actions.
>
> That being said, our current definition provides an initial framework for formalizing strategic relationships, and is still a useful tool as demonstrated by our applications.
>
> ---
>
> **Computational Complexity**
>
> We agree that determining strategic link scores is indeed costly, Algorithm 1 has a complexity of O(TP) as the reviewer has pointed out, and performing Algorithm 1 as it is would not be practical for every decision-making scenario. However, that does not mean strategic link scores are never practical. Algorithm 1 is only one approach we develop using strategic link scores. We give several examples where strategic link scores can be practical despite their complexity.
>
> In Section 5.1, we consider chess, where planning requires reasoning over massive decision trees. But, once we obtain one plan (i.e. the Stockfish solution), reasoning over short game sequences only requires partial reconstructions of the tree, which make our approach feasible.
>
> In Section 5.2. we consider a healthcare scenario, where the complexity of our approach scales with the number of recommendations we want to analyze, $|\mathcal{D}|$, and not with the dimensionality of the environment. In high-stakes domains such as healthcare, the fact that many policy improvement methods aim to find a small set of recommendations to stay in the support of the original policy (Laroche et al. 2019, Wu et al. 2022, Sharma et al. 2024) works in our favor.
>
> In Section 6.1, we consider a traffic scenario, where a single intervention in the form of a road closure allows us to determine multiple strategic link scores and uncover a substantial amount of information.
>
> ---
>
> **Continuous States and Actions**
>
> As the reviewer has pointed out, when the state space or the action space is continuous, removing a single point $(\tilde{s},\tilde{a})$ from the available action set $\mathcal{A}$ in (2) would have practically no impact on the resulting policies. In continuous cases, we must instead consider a region of pay-off decisions $\tilde{\mathcal{S}}\times\tilde{\mathcal{A}}\subseteq\mathcal{S}\times\mathcal{A}$ such that the constrained environment has the action availability: $\mathcal{A}’(s)=\mathcal{A}(s)\setminus\tilde{\mathcal{A}}$ if $s\in\tilde{\mathcal{S}}$ and $\mathcal{A}’(s)=\mathcal{A}(s)$ otherwise.
>
> This is what we have done in our traffic example in Section 6.1, which involves policies that determine traffic flow $a\in\mathcal{A}=[0,1]$ through junctions $s\in\mathcal{S}=$ { $1,\ldots,10$ }. When interested in whether the flow through the first junction $s=1$ is strategically dependent on sufficient flow at the last junction $s=10$, we consider the pay-off region $\tilde{\mathcal{S}}=$ { $10$ } and $\tilde{\mathcal{A}}=(0,1]$. This encodes the constraint that the flow $a\sim\pi(s=10)$ at junction $s=10$ through the arterial road must be zero.

---

> ### Author Response · Authors · 2025-11-23
>
> **Answers to Questions**
>
> 1. Yes, it is the performance with no recommendations implemented (i.e. the value corresponding to zero on the x-axis).
>
> 2. We consider all possible combinations of recommendations and report the average performance and the combination with the worst performance. For instance, for Pick-and-Choose, if the number of recommendations is 5 but the number of recommendations that are implemented is 3, we consider the 5-choose-3 possible ways to implement 3 of the 5 recommended actions (computing both the mean and the worst resulting performance).
>
> 3. Consider the case where the stochasticity is maximal, i.e. the actions are uniformly random. IRL can infer that the reward function must have very small magnitude (corresponding to a high temperature parameter) but cannot identify its shape, which is what our metric, the EPIC distance, evaluates. Any reward structure when scaled sufficiently down would result in an almost uniform distribution of actions. Meanwhile, strategic link scores rely only on accurately modeling action distributions (i.e. policies). Even if the shape of the inferred reward function is incorrect, if it still produces uniformly random policies under all constraints, the strategic link scores between any two actions would correctly be identified as zero.
>
> 4. You are correct, each decision $(s,a)$ is strategically linked to itself with a score of $\pi(s,a)$, not necessarily one.
>
> 5. They show standard variance across five repetitions of the same experiment, as state in Line 779.

---

> > ### Comment · Reviewer_XeLC · 2025-11-28
> >
> > Thank you for the response. My concerns are mostly addressed. Similar to reviewer VG5J, I encourage the authors to clearly discuss limitations and possible remedies in the paper. I have one follow-up question:
> >
> > - For higher-order strategic links, while "how to determine an appropriate level of abstraction remains an open question for future research", can the authors outline some possible ways to achieve this?

---

> ### Author Response · Authors · 2025-12-02
>
> Building on top of the work on option discovery can be one direction towards learning higher level strategies. Maybe one can aim to find option pairs where the use of one option is sensitive to random perturbations made on the other option's actions.

---

### Official Review · Reviewer_VG5J · 2025-10-31

**Soundness:** 2
**Presentation:** 3
**Contribution:** 2
**Rating:** 4
**Confidence:** 3

**Summary:**

The paper proposes a method to expose multi-step strategies in sequential decision-making systems by defining a strategic link score between pairs of actions in different states. A link score $𝑆_{𝑠,𝑎→\tilde{s},\tilde{𝑎}}$ measures how much the policy’s probability of taking an earlier “setup” action $(𝑠, 𝑎)$ drops if a later “payoff” action $(\tilde{s},\tilde{𝑎})$ is made unavailable. Intuitively: “Would the agent still bother doing step 1 if it couldn’t later do step 2?”

The paper argues this reveals strategic dependencies between actions, in a way that reward attribution / value explanations don’t. The paper (1) visualizes these links to interpret behavior in GridWorlds, chess, and traffic and (2) uses them to build “strategy-aware decision support,” where instead of recommending individual actions to a human operator, the system recommends bundles of linked actions so that partial compliance is less dangerous. This is tested in “Shortcuts” and a discretized hypotension treatment environment that mimics ICU decision support.

Finally, the paper claims these links can be estimated even without direct access to the original planner, either by intervening in a simulator (traffic) or via IRL on demonstrations.

**Strengths:**

### 1. Clear, appealing problem statement.
The paper frames a real gap: existing explanations in RL mostly tell me why an action is “good long-term,” but not “this action is only worthwhile if we’ll later do that other action.” The authors focus on action-action contingency, which is intuitive to humans (“I grabbed the key only because I planned to open the door”). This is a crisp interpretability goal.

### 2. Actionable downstream use.
The decision-support angle is compelling. It’s not just interpretability for storytelling; the paper operationalizes it by grouping linked actions into bundles and shows that this “strategy-aware” interface can improve worst-case outcomes when a human only partially follows the agent’s advice, especially in environments with fragile multi-step maneuvers (Shortcuts), and at least does not hurt in a clinically flavored hypotension scenario.

### 3. Breadth of domains.
The method is demonstrated in (i) toy GridWorlds with keys and doors, (ii) a structured traffic network, (iii) a chess sequence with castling and follow-up play, (iv) a simplified ICU hypotension management setup. That breadth helps argue generality and shows the method isn't tied to a single planner or reward function.

**Weaknesses:**

### 1. Novelty vs existing ideas is under-defended.
I am not fully convinced (yet) that “strategic link score” is fundamentally distinct from prior notions like (i) causal influence of future constraints on earlier policy choices, (ii) credit assignment / Q-value sensitivity, (iii) options / skills / subgoal discovery. One might say: “You’re just measuring how much I stop wanting setup action A if payoff action B is impossible,” which sounds like standard counterfactual policy sensitivity. The paper needs a sharper, explicit contrast with causal RL explanations and hierarchical RL. Right now, that contrast is mostly implicit and could be challenged.

### 2. Scalability / practicality.
Computing the link score requires re-planning under counterfactual interventions that forbid specific future actions at specific future states. That’s cheap in tabular / toy planners and in the softmax-over-Stockfish construction, but it’s unclear how this scales to high-dimensional continuous-control agents or giant neural policies without a fast differentiable planner. There is no complexity analysis, runtime table, or approximation method.

### 3. Validation is mostly qualitative.
Evidence that high link scores correspond to “true strategy” is presented as visual heatmaps (e.g. keys↔doors, shortcut pre-commitment, chess piece mobilization) and narrative interpretation. There’s no quantitative benchmark like precision/recall of discovered links against ground-truth enabling relations in GridWorld / Shortcuts, no human annotation study (“does this look like a setup-payoff pair?”), and no false-positive / false-negative analysis. That leaves room for the criticism that this is anecdotal.

### 4. Chess experiment credibility.
The chess case study may look less impressive to experts, because the “policy” is generated by softmaxing Stockfish evals over legal moves rather than reading Stockfish’s true multi-ply principal variation. So it is not obvious that it is revealing latent plans; it might just be reconstructing obvious tactical follow-ups visible from engine eval deltas.

**Questions:**

1. Positioning / novelty. How is the strategic link score not just causal influence / Q-value sensitivity / option precondition discovery? Can you provide (a) a small formal example where traditional credit assignment says two actions are related but your link is zero, and vice versa, and (b) a concise statement of what is genuinely new?

2. Computational tractability. What is the computational complexity of computing all pairwise link scores along a trajectory of length $T$? How many re-plans do you need, and how does this behave for large neural policies where replanning is nontrivial? Do you have an approximation approach (e.g. masking logits in a learned policy network instead of full replan)?

3. Quantitative evaluation. Can you report precision/recall or AUROC for recovering “true” enabling relations (e.g. key→door, shortcut-setup→shortcut-payoff) vs baselines such as (i) temporal proximity, (ii) pure co-occurrence frequency, or (iii) reward-drop ablations? This would turn your qualitative figures into measurable impact.

4. Robustness / false positives. Does a high link score always imply that the setup action is useless without the payoff, or can it also just mean that disabling the payoff collapsed a whole profitable branch of future behaviors that share structure with that payoff? In other words: how specific is S? Are there systematic false positives?

5. Clinical external validity. In Hypotension, the “reward” is a shaped physiological stability score rather than actual patient-centered outcomes, and the human decision-maker is simulated by randomly dropping some actions from the agent’s bundle. How confident are you that this result will generalize to (a) real ICU objectives like mortality and (b) real clinician partial adoption patterns (e.g. “I’ll titrate fluids but I won’t start pressors yet”)? Can you discuss limitations here explicitly, so readers don’t overinterpret?

6. Black-box setting. Section 6 is very interesting. Under what assumptions does the IRL-based reconstruction of strategic links match the ground-truth planner’s links (e.g. identifiability or coverage assumptions)? Right now Fig. 12 suggests performance can improve with more stochastic behavior, but this is only briefly described. Can you elevate and formalize this claim?

---

> ### Author Response · Authors · 2025-11-23
>
> **Novelty**
>
> As we discuss in “Planning-Level XRL”, other methods also explain RL policies by describing how a particular action choice impacts the future rewards an agent will attain (Juozapaitis et al. 2019, Erwig et al. 2018, Madumal et al. 2020, Yau et al. 2020, Cruz et al. 2023). While most work focuses on relating actions to their eventual pay-off (including credit assignment or Q-value sensitivity), we aim to relate actions to necessary follow-up actions.
>
> *Is this a meaningful difference?* Yes! Strategic link scores reveal relationships that existing methods do not capture. For example, in the toy setting of Figure 2, the decision sequence (1st, $\Lambda$) and (2nd, $\Lambda$) is optimal under both the red and blue reward structures. However, strategic link scores reveal that these actions work together under the red reward structure, whereas they are independent improvements over the alternative action $\Omega$ under the blue reward structure. Credit assignment or Q-value sensitivity would not have differentiated the two scenarios because both decisions contribute equally to the final reward, i.e. constraining either (1st, $\Lambda$) or (2nd, $\Lambda$) leads to an equal drop in the optimal reward. Likewise, options/skills/subgoal discovery would not have differentiated the two scenarios either because performing (1st, $\Lambda$) is a precondition for (2nd, $\Lambda$) in both cases (transition dynamics are the same).
>
> *Is this an important difference?* Yes! When recommending actions to an agent, it matters whether our recommendations need to be implemented together (cf. strategic link scores) in addition to how much they will impact the final outcome (cf. credit assignment or Q-value sensitivity). If we recommend Action A and Action B, it is important to know whether Action A enables Action B (cf. options/skills/subgoal discovery) but it is also important to know whether performing Action B is necessary for Action A to be a worthwhile action or whether it is still advantageous to perform Action A on its own (cf. strategic link scores).
>
> ---
>
> **Scalability / Practicality**
>
> Determining strategic link scores is indeed costly, it may not be practical in every decision-making scenario. However, that does not mean strategic link scores do not have any practical applications. We highlight several in our paper.
>
> In Section 5.1, we consider chess, where planning requires reasoning over massive decision trees. But, once we obtain one plan (i.e. the Stockfish solution), reasoning over short game sequences only requires partial reconstructions of the tree, which make our approach feasible.
>
> In Section 5.2. we consider a healthcare scenario, where the complexity of our approach scales with the number of recommendations we want to analyze, $|\mathcal{D}|$, and not with the dimensionality of the environment. In high-stakes domains such as healthcare, the fact that many policy improvement methods aim to find a small set of recommendations to stay in the support of the original policy (Laroche et al. 2019, Wu et al. 2022, Sharma et al. 2024) works in our favor.
>
> In Section 6.1, we consider a traffic scenario, where a single intervention in the form of a road closure allows us to determine multiple strategic link scores and uncover a substantial amount of information.
>
> ---
>
> **Quantitative Validation**
>
> We provide quantitative results. In Section 5.2, we show that using strategic link scores can help deliver measurable improvements over existing decision support strategies (even if we completely ignore whether these scores are interpretable or not, which we assess qualitatively). In Section 6.2, we show that strategic link scores can be estimated from demonstrations with sufficient variation.
>
> ---
>
> **Credibility of Chess Experiments**
>
> We agree that engine evaluations already communicate clearly what are strong follow-ups to a given move. Our contribution is to show that strategic link scores can also reveal whether the strength of a move is dependent on a particular follow-up. We agree that these scores are fundamentally engine evaluation deltas. After all, strategic link scores are defined as policy deltas in (4), and softmaxing is performed to bring engine evaluations (real numbers) to the policy scale (between zero and one). However, our definition requires evaluating board states under restricted move-sets, which is not something Stockfish supports out of the box. This makes the analysis in Figure 5 novel and different from conventional engine-based analysis.

---

> ### Author Response · Authors · 2025-11-23
>
> **Computational Complexity**
>
> Determining the strategic link score between a set-up decision and a pay-off decision requires planning a hypothetical policy under the constraint that the pay-off decision is unavailable. For an RL planner, this means  re-learning a policy from scratch. For non-RL planners, as in our traffic examples, it requires an interventional study. Consequently, when calculating multiple strategic link scores, as in Algorithms 1 and 2, the computational cost (i.e. the number of hypothetical policies we need to plan) grows linearly with the number of pay-off decisions we are interested in. For Algorithm 1, this number is the trajectory length, $O(T)$. For Algorithm 2, it is the number of recommended decisions, $O(|\mathcal{D}|)$.
>
> ---
>
> **Specificity of Strategic Link Scores**
>
> Our definition does not capture all forms of strategic dependence. Some strategies involve higher-order relationships. For example, if A enables either B or C, constraints put just on B or just on C would not reveal the full relationship between A, B, and C.
>
> ---
>
> **Limitations of the Healthcare Scenario**
>
> We will clarify the limitations of the healthcare setup. Having a well-behaved simulation allows us to compute a high-quality policy that improves upon the original policy. What might not hold in practice is the optimality of this improved policy. All methods we compare decide only how to present recommendations, and they all rely on the same improved policy as the source of their recommendations, so they are similarly affected by this limitation. It is worth noting that strategy-aware recommendations are strictly more conservative than the common pick-and-choose approach.
>
> ---
>
> **Inverse Reinforcement Learning**
>
> IRL is accurate when demonstrations are varied enough to provide good coverage of states. In Figure 12, we observe that a similar trend holds for strategic link scores. Since strategic link scores are a direct function of reward functions, it is natural that more accurate reward functions lead to more accurate strategic link scores as well.
>
> However, we had one interesting observation: When the stochasticity of the demonstrator is so high that their actions are essentially random, IRL breaks down (the reward function becomes unidentifiable, as seen in Figures 12a and 12c). Yet the accuracy of strategic link scores remains high. In order to understand why, consider the extreme case where the actions are completely random. Determining correctly that strategic link scores are zero would only require knowing that actions are distributed uniformly rather than pinpointing an exact reward function. In general, the reward function fully characterizes a planner and is therefore the most informative about the planner. Strategic link scores are derived quantities that carry less information about the planner, and hence can be easier to estimate than the reward function itself.

---

> ### Comment · Reviewer_VG5J · 2025-11-24
>
> I thank the authors for clarifying the novelty of their approach. I am inclined to increase my rating. That said, I believe there remain a few things that must be addressed directly in the paper.
>
> 1. Novelty (Addressed): The intuition behind the "Red vs. Blue" example (Figure 2) discussed in the response to my review (and MC5T's review) is rather crucial. It demonstrates that Strategic Link Scores capture contingency in a way that standard Q-value sensitivity or credit assignment cannot. This resolves my primary concern about novelty, but is worth stating more clearly in the paper.
>
> 2. Handling Redundancy (Crucial Limitation): Reviewer XeLC’s critique regarding OR relationships (the GridWorld scenario where "instead of having a single door, there are two neighboring doors unlocked by the same key") is valid. This limitation must be explained in the paper. As a reader, it is important to know that the Strategic Link Score captures _necessary, specific dependencies_ and may result in false negatives in cases of redundancy.
>
> 3. Complexity: I accept the authors' position that for high-stakes, offline decision-making (e.g., healthcare), the O(TP) complexity when assuming a planner is acceptable. The Strategy-Aware downstream performance (Figure 7) serves as sufficient indirect validation of the metric's utility. (Nit: please use "Strategy-Aware" consistently instead of "Strategic-Aware".)
>
> Conclusion: The authors have clarified the method's uniqueness relative to baselines. Provided they (a) revise Section 4 to explicitly contrast their method with Q-values using the "Red vs. Blue" example, and (b) add a clear "Limitations" section discussing both the $O(TP)$ complexity and the failure to capture redundant ("OR") strategies, I will raise my score to 6.

---

### Official Review · Reviewer_9T6i · 2025-11-01

**Soundness:** 2
**Presentation:** 3
**Contribution:** 2
**Rating:** 4
**Confidence:** 3

**Summary:**

This paper introduces a novel interpretability metric, "strategic link scores," designed to quantify dependencies between actions in policies, $\pi$, derived from long-horizon reinforcement learning. The score is formally defined as the decrease in the likelihood of selecting an "enabling" action, $\pi(a|s)$, when a related, future "follow-up" action, $\tilde{a}$, is unavailable. The authors motivate this by noting that optimal policies, $\pi^*$, often involve short-term sacrifices (i.e., selecting $a$ where $r(s, a)$ is low) to enable long-term gains. The utility of this score is demonstrated through policy analysis in several tabular planning environments and within the Maximum Entropy (MaxEnt) Inverse Reinforcement Learning (IRL) framework.

**Strengths:**

The paper addresses the critical and challenging problem of policy interpretability in long-horizon sequential decision-making. The idea of quantifying the link between a "setup" action ($a$) and a "payoff" action ($\tilde{a}$) is an intuitive and valuable contribution to the field of Explainable RL (XRL).

The core contribution, the "strategic link score," is clearly and formally defined. This provides a concrete, computable metric for what is often not a very clear, qualitative discussion of "strategy."

The authors connect their proposed metric to an established field by showing how strategic link scores can be used to analyze the results of Maximum Entropy (MaxEnt) IRL, providing a practical use case.

**Weaknesses:**

The paper is not very clear on the computational expense of the calculation of the strategic link score. Can the authors elaborate on the computation cost and the increase in the complexity analysis? Possibly in $\mathcal{O}(|\mathcal{S}|, |\mathcal{A}|)$

I believe that the definition of set-up action and pay-off is not very rigorous and requires to be more mathematically grounded in order to increase the applicability of the method to more applications. Additionally, this definition should connect the actions clearly to the defined MDP.

Moreover, the limitations of the paper are not clearly stated. I believe the inclusion of the limitations would further improve the readability of the manuscript.

**Questions:**

Authors should include at least two baselines (state-of-the-art) to verify their methods against them; if such methods do not exist, it should be clearly stated.

The manuscript mentions that inverse reinforcement learning (IRL) is used for strategic link scores using reward-based modeling, but it is not clear whether the strategic link score is used as an analysis tool applied to the results of IRL or whether it is integrated in the maximum entropy IRL objective function.

The paper's core objective is to analyze the cause-and-effect relationship between a setup action ($a$) and a future payoff action ($\tilde{a}$). This raises the causal relation among the actions and is fundamentally a question of causal influence. However, the manuscript fails to connect this work to the well-established and highly relevant field of Causal Reinforcement Learning [2], [3]. The authors should position their work within this literature, compare their proposed score to existing methods in Causal RL for identifying action influence, and justify why a new metric is needed.

[1] Bertoin, David, Adil Zouitine, Mehdi Zouitine, and Emmanuel Rachelson. "Look where you look! Saliency-guided Q-networks for generalization in visual Reinforcement Learning." Advances in neural information processing systems 35 (2022): 30693-30706.
[2] Seitzer, Maximilian, Bernhard Schölkopf, and Georg Martius. "Causal influence detection for improving efficiency in reinforcement learning." Advances in Neural Information Processing Systems 34 (2021): 22905-22918.
[3] Madumal, Prashan, Tim Miller, Liz Sonenberg, and Frank Vetere. "Explainable reinforcement learning through a causal lens." In Proceedings of the AAAI conference on artificial intelligence, vol. 34, no. 03, pp. 2493-2500. 2020.

---

> ### Author Response · Authors · 2025-11-23
>
> **Computational Complexity**
>
> Determining the strategic link score between a set-up decision and a pay-off decision requires planning a hypothetical policy under the constraint that the pay-off decision is unavailable. For an RL planner, this means  re-learning a policy from scratch. For non-RL planners, as in our traffic examples, it requires an interventional study. Consequently, when calculating multiple strategic link scores, as in Algorithms 1 and 2, the computational cost (i.e. the number of hypothetical policies we need to plan) grows linearly with the number of pay-off decisions we are interested in. For Algorithm 1, this number is the trajectory length, $O(T)$. For Algorithm 2, it is the number of recommended decisions, $O(|\mathcal{D}|)$.
>
> ---
>
> **Definition of Set-up and Pay-off Decisions**
>
> The strategic link score between two decisions $(s,a)$ and $(\tilde{s},\tilde{a})$ is defined in (4). We refer to the former as the set-up decision and the latter as the pay-off decision. This is because a high score indicates that $(s,a)$ acts as the setup for the payoff of $(\tilde{s},\tilde{a})$. Following our definition in (4), the policies $\pi$ and $\pi\_{\neg\tilde{a}|\tilde{s}}$ correspond to $\mathcal{P}(\mathcal{E})$ and $\mathcal{P}(\mathcal{E}\_{\neg\tilde{a}|\tilde{s}})$ respectively (as stated on the same line). The planner $\mathcal{P}$ is an arbitrary planner introduced in Line 130, with the definition of a reward-based planner as an example in (1). Finally, the MDP $\mathcal{E}$ is introduced in Line 121, and its constrained variant $\mathcal{E}\_{\neg\tilde{a}|\tilde{s}}$ is defined in (2).
>
> ---
>
> **Limitations**
>
> *Continuous actions.* When the state space or the action space is continuous, the definition of strategic link scores becomes problematic: Removing a single point $(\tilde{s},\tilde{a})$ from the available action set $\mathcal{A}$ in (2) would have practically no impact on the resulting policies. In continuous cases, we must instead consider a region of pay-off decisions $\tilde{\mathcal{S}}\times\tilde{\mathcal{A}}\subseteq\mathcal{S}\times\mathcal{A}$ such that the constrained environment has the action availability: $\mathcal{A}’(s)=\mathcal{A}(s)\setminus\tilde{\mathcal{A}}$ if $s\in\tilde{\mathcal{S}}$ and $\mathcal{A}’(s)=\mathcal{A}(s)$ otherwise.
>
> A meaningful specification of pay-off regions is application-dependent. For example, our traffic scenario involves policies that determine traffic flow $a\in\mathcal{A}=[0,1]$ through junctions $s\in\mathcal{S}=$ { $1,\ldots,10$ }. When interested in whether the flow through the first junction $s=1$ is strategically dependent on sufficient flow at the last junction $s=10$, we consider the pay-off region $\tilde{\mathcal{S}}=$ { $10$ } and $\tilde{\mathcal{A}}=(\ell,1]$ for some threshold $\ell$. This encodes the constraint that the flow $a\sim\pi(s=10)$ at junction $s=10$ cannot be larger than the threshold (i.e. $a\leq \ell$).
>
> *Higher-order strategic links.* Our definition does not capture all forms of strategic dependence. Some strategies involve higher-order relationships. For example, if A enables either B or C, constraints put just on B or just on C would not reveal the full relationship between A, B, and C.
>
> ---
>
> **Baselines**
>
> We propose a new measure, the strategic link score, and our primary interest is in its utility rather than the efficiency or accuracy of its computation. As evidence for this utility, we consider multiple applications, and when appropriate, we compare any new methods enabled by strategic link scores to existing approaches. For instance, in Section 5.2, strategic link scores enable the strategy-aware recommendations in Algorithm 2, and the results in Figure 7 compare this approach to the common “Pick-and-Choose” baseline used in decision support systems, where actions are recommended on a state-by-state basis as they are encountered.

---

> ### Author Response · Authors · 2025-11-23
>
> **Inverse Reinforcement Learning**
>
> Calculating a strategic link score requires the following inputs: (i) an environment $\mathcal{E}$, (ii) a planner $\mathcal{P}$, and (iii) two decisions $(s,a)$ and $(\tilde{s},\tilde{a})$. Which decisions we examine depends on the application (for instance, whether we look at interpretability or recommendations). Throughout the paper, we assume that the environment is known (i.e. we can at least sample from it). In the applications in Section 5.1 and 5.2, we assume that the planner is also known (i.e. we can provide an arbitrary environment and obtain a policy for that environment in return). In Section 6.2, we consider what to do when we do not have access to the planner. IRL provides one solution: It takes demonstrations as input and outputs a reward function that is estimated to be optimized by the demonstrator. This recovered reward function models a planner, i.e. the planner that optimizes for that reward function. We use this planner when calculating strategic link scores, treating it as a known planner as before and disregarding demonstrations from then on.
>
> ---
>
> **Relationship to Causal RL**
>
> Some work in Causal RL is interested in finding causal relationships in environment variables / transition dynamics for efficient RL (for instance, [2] falls into this category). Our goal is not to perform RL. Rather, we are interested in explaining decision-making behavior (which may be the behavior of an RL agent). Other work uses causal relationships to explain policies. [3] is indeed an example of this and we already cite it in our paper in Line 107, where we discuss how the explanations provided by [3] (and other related work) differs from those produced by our methodology.

---

### Meta-Review · Area_Chair_exUY · 2026-01-12

**Summary:**

This paper proposes a novel metric to do interpretable RL called _strategic link scores_. This score computes score between one state-action pair, known as the setup decision, and another called pay-off decision by seeing the probability by which the learned policy would have changed in value for the setup decision, if the pay-off decision was unavailable. Proposed applications are explaining link between decisions, and recommending actions for improving performance.

Main concerns unaddressed here were computational efficiency, higher-order interactions, and better positioning in the related work. The first two concerns may require changing the approach quite a bit. Specially, the computational efficiency makes the approach not suitable for complex real-world problems. For these reasons, I am going with a weak reject.

**Reviewer Concerns:**

1. Reviewer 9T6i mains concerns were computational efficiency, lack of baselines, and comparing the work with Causal RL literature. The main concern here that remains is that the proposed approach scales with the number of pay-off decision which can be an expensive application. Further, what happens if the planner is stochastic, in which case one may have to do multiple replicates and average.

2. Reviewer VG5J raised concerns about lack of comparison with other similar metrics (e.g., causal influence), practical efficiency, and lack of quantitative analysis. The main concern here that remains is expense of computing strategic link scores.

3. Reviewer XeLC raised concerns on higher-order interaction and computational complexity both of which remain.

4. Reviewer MC5T raised concerns on soundness and novelty of the given approach. I believe authors have partially addressed these concerns specially on difference from past work. However, they noted that the problem of right abstraction remains open.

**Reviewer Scores:**

1. Reviewer 9T6i may have increased their score to 6.

2. Reviewer VG5J mentioned they'd increase their score to 6, if authors make some changes in their writing which I am hoping they would. Therefore, I'd believe Reviewer VG5J will increase their score to 6.

3. Reviewer XeLC would have likely kept their score at 4.

4. Reviewer MC5T may have increased their score to 4.

---

### Decision · Program_Chairs · 2026-01-26

Reject